# Ambient Hot Box: An Instrument for Thermal Characterization of Building Elements and Constructive Materials

**DOI:** 10.3390/s23031576

**Published:** 2023-02-01

**Authors:** Cristian Carmona, Joan Muñoz, Bartomeu Alorda-Ladaria

**Affiliations:** 1Department of Industrial Engineering and Construction, University of the Balearic Islands, 07122 Palma, Spain; 2e-Health and Multidisciplinary Telemedicine Research Group, Institute for Health Research Illes Balears (IDISBA), 07010 Palma, Spain; 3Agro-Environmental and Water Economics Institute (INAGEA), 07122 Palma, Spain

**Keywords:** hot-box, fast thermal characterization, thermal isolation, energy efficiency in buildings

## Abstract

In assessing the energy performance of buildings, the thermal performance of the structural components and building materials is crucial. Although reference catalogs are used to determine the thermal properties of construction materials, the use of novel materials or non-homogeneous mixtures, particularly with biomaterials, demands the development of new instruments that are capable of performing rapid, accurate and cost-effective thermal characterization. This study introduces the ambient hot-box, a new tool for measuring the thermal properties of construction components and heterogeneous materials. The paper provides a methodology for measuring a sample’s benchmark and fresh materials using a streamlined hot-box-based instrument. Utilizing samples as a benchmark material, the new instrument is assessed, yielding transmittance values with errors below 4%. The electronic circuits, measurements techniques and instrument implementation are all described.

## 1. Introduction

In a context of climate change and energy cost increments, one of the largest energy waste challenges in today’s smart and sustainable cities models are related to reducing the energy demand of buildings. Specifically for the EU-28 countries, according to reports published by the Directorate General for Energy of the European Commission [1], approximately 30% of the energy consumed is due to residential buildings and an additional 30% is due to commercial consumers and utilities. Thus, the need to improve energy management in buildings to enable a sustainable future seems clear [2,3].

A building’s energy demand is influenced by several factors [3], including the envelope’s thermal performance, climatic conditions, the efficiency of heating/cooling systems or electrical appliances, and user behavior [4].

Among these, the building envelope’s insulation performance has a significant impact on the total energy performance of the building and a near zero energy strategy is growing to overcome this situation [5]. Energy demand related to building construction elements depends mainly on the following factors: (1) the building envelope [6,7,8] (roofs, floors and walls); 25% of heat loss in buildings comes from attics, 15% from basements/floors, and 35% from walls; (2) openings in the building envelope [9,10] (doors and windows) 25% of windows/doors; and (3) thermal bridges [11,12] (wall-to-wall, wall-to-door, wall-to-window and wall-to-floor junctions). In this sense, it can be stated that thermal bridges in buildings reduce the effectiveness of thermal insulation by 40%. Therefore, the characterization of the actual thermal parameters of building envelopes has become one of the main concerns of building engineers when it comes to improving energy efficiency in construction projects.

The thermal performance of structural elements and materials are continuously analyzed and improved to reduce heat losses and increase the efficiency of building projects [13]. The thermal transmittance value (U-value), which measures how well insulated a building element is, provides an accurate assessment. The material’s capacity for insulation improves with decreasing U-value. The resistance values of each individual material that makes up a building element are traditionally added to determine the U-value. In this sense, the thermal characterization of a wide variety of construction materials is reported in official catalogs; this is the case for Spain [14]. This catalog-based methodology is useful in estimating the energy performance of a building when the materials are known and included in the catalog. However, they are barely applicable in circumstances where either the material data are unknown or the structure of the building elements are highly variable, such as in restoration or refurbishment projects, with variances of up to 50% [15]. In this sense, several previous works have reported significant variability in thermal resistance estimation of up to 30% depending on the methodology used [16].

In addition, the use of natural materials, such as straw or masonry walls, are clear examples of materials with high transmittance variability due to their intrinsic heterogeneity. In general, this circumstance is solved by applying a thermal transmittance average value. Although, this straightforward assumption is useful as an initial approximation, the heterogenous structure of the natural material may substantially modify the thermal transmittance final value [17].

Other construction elements that present this problem are those based on composite materials, such as mud-straw compressed blocks, where even if the proportions of the material remain the same, small physical variations in the different material used (humidity, compression, etc.) lead to significant variations in the final thermal transmittance value [18]. Experimental analysis must be used to resolve these difficult problems and help improve building energy efficiency. In this sense, the terms destructive and non-destructive methodologies can be used interchangeably [19]. When using destructive methodology, a portion of the insulation envelope is destroyed to obtain the experimental parameters for integrating the sensors and automatic monitoring systems, while the constructive element is not altered during the non-destructive methodologies’ analysis. In these last methodologies, a number of methods have been put forth and contrasted [20,21] to accurately determine the U-value based on simple hot-box heat-flux meters [22,23], infrared thermography [24,25,26], temperature-based methods [27], temperature control-box heat-flux meters [28] or heat-flux meters [29,30]. Although using in-situ non-destructive techniques can provide a good adaptation for complex envelopments, most destructive projects may encounter challenges due to the necessity of ongoing monitoring. Therefore, experimental approaches that focus on lab setup are preferred.

The method based on hot-box methodology were proposed as international standard in ISO 8990:1994 [31] and ASTM C1363-19 [32], because it recreates real and repeatable thermal conditions and is the most used thermal characterization methodology in construction systems [33]. The characterization process is essentially based on placing the material to be studied between two different temperature chambers called hot and cold boxes. A completely stable state is created to measure different experimental parameters and obtain the under-study material’s thermal transmission properties. The hot-box methodology proposes to extract the thermal features of a material measuring: the temperature variables from one site to the other site of the material; the thermal energy flow that circulates through the sample [34]; and the amount of energy that must be supplied to maintain the steady state [35]. The main drawback of this methodology is related to the time needed to achieve a completely steady state in an inhomogeneous sample with variability of humidity/density in the material. In addition, it is necessary to maintain the sample at constant temperature for two weeks before the test can be performed [34,35]. The cost of constructing a complete hot-box apparatus is another important factor limiting the use of this type of methodology for accurate thermal characterization. This is particularly dramatic in renovation projects where the thermal characteristics are not fully understood and an exhaustive test cannot be conducted due to the high costs associated with obtaining such certifications in terms of money and time.

Finally, the ISO 9869 standard [36] and ASTM C 1155 [37] describes a thermal resistance estimation using an in-situ measurement methodology. Both methods measure the thermal resistance of a building element using two thermocouples fixed opposite to each other on both sites of the wall. The heat-flux sensor is added near to the thermocouple position of one site. During the test, the interior side will preferably offer higher stability in temperature. The main drawbacks are, firstly, that the data obtained must meet the criteria of standards, then measurement duration can be very long due to low-temperature differences between both wall sides [13,38], and secondly, reported data must be corrected with information about heat storage and measuring conditions to improve the estimation accuracy [38,39]. Although the in-situ-based measurements are easy to obtain especially if a heat flux meter is used [40], the measurements should be analyzed by a person with experience in order to obtain reliable results [39], especially when the measurement is performed in a low heat-flux regime [40].

The aim of this paper is to propose an electronic instrumentation system that allows obtaining a thermal resistance value of a building element, either a simple material or a complex building element, in a simple, appropriate duration and cost-effective way. The proposed apparatus simplifies hot-box instruments to offer:Accurate and fast thermal resistance estimation;Adaptability to regular and irregular samples with fast and low-cost rebuilding of the proposed hot box;Fulfilling the existing constructive material characterization regulation;Reducing measurement setup complexity maintaining estimation error below the maximum allowed by regulation.

The rest of the paper is organized in the following sections: Section 2 describes the main hot-box structures described by international standards, the new ambient hot-box (aHB) instrument and the methodology to obtain the data. Section 3, reports the results of the new apparatus used on benchmark and non-homogeneous materials. In this section, the maximum error in the thermal parameter estimation is evaluated to validate the new instrument. Finally, the discussion and conclusions are pointed out in Section 4.

## 2. Materials and Methods

The thermal characteristics of a building element are mainly defined by their thermal transmittance W/m^2^·K and thermal mass J/K, although current legislation and regulations pay special attention only to the transmittance of enclosing wall elements. Thermal transmittance (U-value) is defined as the amount of heat flow that can pass through a surface per unit of time, when that surface is exposed to a thermal gradient of 1 K, Equation (1).
(1)$$U\text{-}\mathrm{value}=\frac{\mathrm{heat}\;\mathrm{flux}}{\Delta \mathrm{Temp}}$$

The operation of the hot-box instrumentation system is based on two principles, a first one established by EN ISO 8990 [31] and ASTM C1363-19 [32] standards in which a sample is exposed to two predetermined temperatures. The thermal resistance of the sample is established by the amount of energy required to keep the two temperatures constant. The second principle is based and defined by GOST 26602.1-99 [41], where the sample is exposed to two stable temperatures, although, in this case, the thermal flux transmitted through the sample itself is measured to obtain the thermal behavior of the sample.

Table 1 summarizes the list of standards related to the aHB design. The table includes its significance with respect to this work and the measurement of the thermal resistance of materials.

The main parts of a hot-box instrument are defined in the different methodology standards presenting some similarities summarized in Figure 1. A hot-box is formed by two main zones: the metering chamber, or hot zone, where the sample is influenced by a constant thermal flow and represents the indoor ambient in a building application. And the climatic chamber, see Figure 1, where the sample transmits the thermal energy changing the chamber ambient. The energy transmission can be accelerated or maintained over time creating a colder ambient which maintains the climatic chamber at lower temperature than the metering chamber. For the correct estimation of the amount of energy that is effectively transmitted through the sample, in relation to the thermal losses, it is necessary to measure the different energy wastes depending on the hot-box methodology used. That is, the losses through the box itself or additionally, transmission losses of the material to external environments should be considered to obtain the most accurate thermal resistance estimation. In order to create a hot-box system and measure the thermal performance of building materials and construction systems, a hot-box must meet calibration and requirement standards. This calibration is described in three main sections: the system itself (hot-box assembly calibration); its components (sensor calibration); and the procedures to be followed to achieve the required reliability.

This paper proposes a simplification of the calibrated and hot plate hot-box methodology and analyzes the advantages and disadvantages of the proposed aHB. The simplified aHB uses only one stable and controlled temperature source to define the temperature in one side of the sample, while leaving an uncontrolled temperature source, ambient temperature, in the other sample side. In this way, both the cost and the complexity of the measuring instrument are substantially reduced obtaining, by contrast, the thermal resistance estimation with enough accuracy for building applications.

### 2.1. The Proposed Instrument

The new aHB instrument introduces the sample under test in the metering chamber. The main difference between a hot-box standard design and the proposed aHB instrument is the absence of a climatic chamber. The main goal of a climatic chamber is to create an ambient environment with stable temperature and very reduced wind speed offering a completely stable steady state, Figure 1. On the contrary, the aHB instrument does not have its own climatic chamber, so we do not have an exhaustive control of the external temperature, nor is the air velocity over the measurement surface substantially controlled. Therefore, a complete steady state cannot be achieved. In that case, the aHB instrument approximates the energy stability of the sample as a function of the thermal flow circulating through it due to both thermal variations produced with respect to the ambient environment and the losses associated with the non-controlled environment. In the next sections the thermal measurement principles and the aHB instrument are described in detail.

#### 2.1.1. Thermal Characterization Principles

Using a thermal gradient between the sample’s faces and the methodology outlined in ISO 8302 [42], it is possible to estimate a material’s thermal resistivity while minimizing losses. Using this standard methodology, the thermal resistivity estimation can be achieved using direct thermal flux measurement and controlling the rest of thermal sources or losses.

The aHB instrument does not have a cold focus of known value (climatic chamber), using instead the surrounding environment to dissipate the heat from the material site. This difference reduces the time required to stabilize the cold focus, the energy required, and the equipment involved. Since the cold temperature cannot be available beforehand, there is no previous information on the thermal characteristics and range of magnitudes of the thermal gradient between the faces of the sample under study. Therefore, it is necessary to continuously measure all thermal variables (temperatures on both exposed faces, heat flow though the sample and temperature losses) from the pre-heat initial conditions and while the hot source is creating the thermal gradient.

The new aHB should minimize the impact of thermal losses, so it is necessary to characterize all error sources present during the test. Figure 2 shows the possible thermal paths that the heat flux can take during the test. The direct heat flux generated by the hot plate is injected into the sample during the test. While *Q* is the heat flux measured on the opposite face in contact with ambient environment, *Q_FL_* are the paths of flanking heat-flux losses and *Q_L_* are the paths of lateral heat-flux losses.

Ideally, without heat-flux losses, the value *Q* will be equal to the heat flux induced by the hot plate. But in a real setup, the value of *Q_FL_* is equal to the flowing heat flux though the insulating material used to adapt the instrument to different sample sizes and shapes, and the *Q_L_* value is equal to the flowing heat flux though the lateral and lower instrument walls. In this sense, minimizing the values of *Q_FL_* and *Q_L_*, the *Q* value will remain stable over time and equal to the heat flux generated by the hot plate. When this condition happens, the sample is thermally stable, and the thermal transmittance can be obtained using Equation (1). Figure 3 shows a schematic representation of U-value evolution when heat losses are minimal, and the sample is exposed to a temperature gradient incrementally. From initial state, the sample under test moves to stable state passing through a transient state, while the hot plate is working, injecting thermal energy to the sample. During transient period, the heat flux, and the temperature gradient change over time. The continuous monitoring of temperature and heat flux variables helps to determine the instant where the sample enters into the stable state. This condition is established when the U-value slope is lower than 5%. In that case, the stable U-value can be defined as the final thermal transmittance value of the material.

This thermal characterization procedure is applied in this work, where a real aHB instrument is constructed and a control electronic circuit is implemented to automatically measure the U-value using a calibrated aHB instrument, where both the sample thermal conditioning time and the complexity of the whole instrument are reduced compared with existing commercial equipment.

In the next section, the ambient hot-box implementation and the electronic circuit are described in detail.

#### 2.1.2. Ambient Hot-Box Description

The aHB instrument consists of a well-calibrated metering chamber formed by insulating walls that isolate the interior space from the outside environment. Figure 4 shows a detailed diagram of the proposed aHB instrument. The insulating walls or external insulator (ExIn) are made of expanded polystyrene (EPS) forming a top-open box to create the metering chamber. The heat flux losses through the lower wall are monitored using a temperature sensor (T_HP_–temperature of hot plate) installed in the lower part of the housing. The hot plate (HP), which acts as a heat source and is positioned over this sensor, creates a temperature gradient between the two sides of the sample. Above the HP a metal guard (MG) is installed to isolate the sample from the HP and reduce contact thermal conductivity between the sample under test and the HP. The hot temperature (T_H_) sensor is located on top of the MG layer and under the sample. The T_H_ sensor monitors the temperature on the hot side of the sample under test.

When the sample material or element is in place on the TH sensor, the space between the sample and the ExIn walls is filled with backfill material. This backfill material (EPSIn–EPS insulator in Figure 4) will remain in contact with the sample under test and provide the best padding to minimize lateral and flanking heat losses. In the proposed aHB, the sample under test is surrounded with expanded polystyrene in the form of 3 mm beads with a density of 40 kg/m^3^. In this way, the width of the insulation (ExIn + EPSIn) between the sample and the external ambient is increased while better adaptation to the specific shape and surface irregularities of the sample is achieved.

In addition, the EPSIn material allows the use of a relatively small dimension of samples, minimizing lateral losses. In standard hot-box instruments, the minimum measurement area is specified as three times the sample width or at least 1 m^2^ when the minimum sample size is 1.5 m. In the aHB case, considering the small size of the instrument, the measurement area is bounded by the diameter of the circle between S_min_ and S_max_, see Figure 4. The S_min_ value is defined by the diameter of the circle covered by the heat flux sensor, while the S_max_ value is defined by the maxim diameter of the circle of the metering chamber. It has been observed during experiments that it is possible to reduce the sample by at least three times the EPSIn thickness considering a wide range of sample shapes.

Finally, only the upper side of the test sample should be exposed to the outside air before the EPSIn material is removed.

The aHB instrument is equipped with three sensors installed in the upper side: a thermal flux sensor (FS–Flux sensor) with a temperature sensor (T_C_–cold temperature) is collocated in the upper side of the sample. An additional temperature sensor is located directly above the EPSIn material to estimate the flanking temperature losses (T_FL_). Figure 5a shows the lateral ExIn walls of the aHB instrument implemented where the upper open side of metering chamber can be observed. Figure 5b shows the final disposition of the heat-flux sensor and EPSIn balls covering the sample under test.

The proposed aHB instrument is composed of 4 temperature sensors, 1 heat flux sensor, 1 hot plate and a metal guard, and the insulation materials to create the metering chamber. All electronic sensors and heaters are controlled with a specific electronic system described in the next section.

#### 2.1.3. Electronic System Description

The thermal gradient is created using the HP, Figure 4. The HP consists of a supply power controlled digitally and a resistive square plate (220 mm × 220 mm × 3 mm) able to operate at 250 Watts. The metal guard (MG in Figure 4) consists of a 5 mm aluminum plate. The presence of the MG avoids the creation of hot spots, minimizes the temperature gradient between different HP points and reduces interferences between HP and the electronic sensors.

The aHB instrument is monitored using temperature sensors and a heat flux sensor. The temperature sensors selected were based on NTC thermistors (NTCLG100E2104JB) with glass encapsulated, 100 KOhm of resistance value at 25 °C, tolerance of ±5%, and quick response time (<0.9 s). The heat-flux sensor selected was the HFP01-05 from Hukseflux with a wide measurement range (−2000 to 2000 W/m^2^), sensing area of 0.0008 m^2^, sensitivity nominal of 60 mV/(W/m^2^), and uncertainty of calibration of ±3%.

The aHB sensors are managed using a microprocessor-based electronic digital system. The electronic signals from the sensors are amplified, filtered, and adjusted in a specific adaptation stage, which deliver the final signals for digital conversion using a MEGA Arduino board. The converted digital data are stored in a Raspberry Pi board using the USB virtual port connection. The final user can access the final acquired data using the remote access to the storage system through an ethernet-based standard connection. A schematic block diagram of the whole electronic system formed by amplifier stages, filter stages (high-pass filter and low-pass filter), adaptation circuits, ADC, microcontroller, and data-collected server is presented in Figure 6.

The FS signal is pre-amplified using an instrumentation amplifier circuit with a fixed amplification factor (G = 10). The pre-amplification stage adapts noisy FS signal to the next filter stages. The filter stages minimize the electromagnetic noise produced by the HP element and power supply. The next step adds a DC offset voltage (2.5 DCV), shifting the analog input signal to the midpoint of the voltage range of the A/D converter (5 VDC). Finally, the last amplification stage adapts the signal to maximize the total input range of the ADC. A variable-gain buffer amplifier (from G = 150 to G = 750) is used to adapt the signal acquisition to different sample materials, maximizing the final FS signal precision.

The NTC temperature sensors are connected to the ADC inputs using the typical adaptative stage based on voltage divider schema and using the manufacturer coefficients and circuit design recommendations.

The final aHB implementation is shown in Figure 7. All sensors from the hot-box are connected directly to the electronic system using the adaptation stages described previously. The microcontroller board manages the signal acquisition and the thermal test methodology to perform the U-value estimation. The next section will describe this test methodology and the error sources and calibration analysis.

## 3. Results

A new aHB test algorithm is proposed to accurately estimate the U-value. Although the temporal requirements of the aHB test are not as important as the final U-value precision, there are some time-related features which should be observed: the ambient and the inside temperatures change slowly (typically 0.5 °C/s) and the heat flux has similar variability (less than 50 W/m^2^·s). Therefore, the ADC sampling frequency is defined around 10 Hz to achieve enough accuracy values. This sampling frequency is easily accessible for most embedded ADCs in embedded microcontroller systems.

Additionally, the duration of test periods (transient and stable in Figure 3) is related to different issues:Thermal absorption features of sample under testInitial HP temperatureInitial sample temperature and moistureAmbient temperatureThermal gradient defined for the testSample sizeLateral heat losses

Considering these timing restrictions and the thermal principles of the aHB, the next subsection describes the proposed measurement methodology.

### 3.1. Obtaining the Final U-Value

The aHB test algorithm provides continuous data measurement during transient and stable periods. The initial temperatures of sample, instrument and ambient are important to reduce the test duration, although they produce marginal impact on the final U-value precision if the operating temperature range of instrument is respected.

The transient period starts at ambient temperature and ends when the temperature arrives at the stable test temperature. This stable point is established when the variability of the Tc sensor data remains lower than 5% with a minimum experiment time of 48 h. This maximum time has been established according that the construction component with the highest thermal capacity and, therefore, that will require longer stabilization times of the temperature flow, will be the aggregates. In this sense, although the time for stabilization will depend on the materials of the sample, considering the aggregates’ material to define the maximum measurement period, maximizes the applicability of the instrument for the construction sector. Additionally, during all the stabilizing process, the sensor data are sampled by the digital circuit and stored in the server database for post-processing purposes.

The thermal resistance value may depend on the temperature difference range applied to the sample under study [43]. The aHB has been designed as a building-oriented instrument, so, the highest gradient is established at 45 °C. This temperature gradient is considered the worst foreseeable case between the inside and outside of a house at Mediterranean latitudes. However, the instrument can apply higher temperature gradients since the HP can reach up to 90 °C without affecting the integrity of the instrument.

The final U-value is estimated applying Equation (1) during the stable period. The rest of sensor data stored during the test could be important for estimating the thermal behavior of the constructive element, because these data could be related to the thermal capacity of the materials forming the constructive element and may be considered as inertial mass in thermal comfort studies in building projects.

Read initial temperature values

While (not STOP Command)

{  Power on HP;

if (Tc < GradientTemp) then

{  read and store sensors data;}

if (Tc = GradientTemp) then

{  Maintain Tc value; }

}

Power off HP

While (not Initial temperatures)

{ read and store sensors data;}

The hot-plate power-on process is implemented using a switching power transistor configured like a motor driver due to the inductive component of the heater.

### 3.2. Error Sources and Instrument Calibration

Although there are numerous modifications in the form and use of the hot-box element itself, in relation to the description of the standard, the physical and material characteristics of the system described are still known. If the details and materials that compose the hot box are known, it is possible to establish, by corrective calculations, the magnitudes of losses produced by the non-ideality of the system itself.

The calibration of hot-box instruments based on standards such as [31,32], is defined using a thermal balance analysis. That is, the total heat energy introduced to the system (*Q_IN_*) should be equal to the energy transmitted through the sample (*Q* in Figure 2), the lateral energy losses (*Q_L_* in Figure 2) and the flanking energy losses through the insulate material in contact with the sample (*Q_FL_* in Figure 2). Equation (2) describes this thermal balance where *Q_IN_* is the heat-flux injected by the HP, *Q_CHW_* is the heat transfer through the bottom of the aHB instrument, and the rest of thermal fluxes defined in Figure 2.
(2)$$Q_{IN}=Q+Q_{CHW}+Q_{L}+Q_{FL}$$

The calibration methodology described in the standard [32] establishes the requirements and parameters to be evaluated for new instruments: a calibration of the hot-box assembly; a calibration of the monitoring system using sensors; and a calculation of the uncertainty of the results. The calibration of the aHB walls has been estimated measuring the heat flux through the different sides of the box in comparison with the total heat flux generated. Table 2 summarizes the percentage of heat-flux losses considering different measuring locations with respect to the HP position. The percentage of losses at the bottom is the highest due to its proximity to the HP. Taking measurements at different heights on the lateral walls, the heat loss profile can be estimated when approaching to the upper side of the aHB. As is expected, the highest percentage of losses are near the HP location while the heat flux losses decrease away from the HP.

Thermal losses due to heat transfer through the bottom side of the aHB (*Q_CHW_*) can be substantially reduced increasing the EPS width by covering the part of the EPS near the HP and exposed to the ambient with vacuum-insulated panels like the ones used for instrument walls.

On the other hand, the flanking losses Q_FL_ can also be reduced by using expanded polystyrene balls with a smaller dimension, increasing the material compaction, and the density, with balls with a maximum density of 640 kg/m^3^ being available on the market.

Following the hot-box calibration methodology, the cables, connectors, and protections used in the new aHB were measured using calibrated instrumentation to estimate their parasitic influence on the analog signals. The acquisition process was adjusted considering these parasitic impacts to provide accurate sensor data.

With the information about the losses and electronic equipment tolerances, several experiments are reported on in the next section showing the principal features of the new aHB instrument proposed.

### 3.3. Material Evaluation Results

The calibration procedure was performed applying the test methodology on a well-known material. The pre-calibrated sample was the Sylvactis 110 SD. A constructive block of 1200 mm × 600 mm × 60 mm was formed of wood fibers with the following thermal features:Thermal conductivity 0.039 W/m·K;Density 110 kg/m^3^;Mass thermal capacity 2000 J/kg·K.

The initial experiment analyzed the increment of lateral heat-flux losses as a function of the temperature gradient created on the sample. Different temperature gradients (from 20 °C to 60 °C) were used to measure the lateral losses. Figure 8 shows the values measured at the middle part of a lateral wall. As was expected the lateral losses increased with the temperature gradient. The linearity of this increment is demonstrated in Figure 8 with the near to 1 value of linear coefficient (R^2^).

In the next experiments, the maximum temperature gradient was established between 30 °C and 50 °C due to its linearity behavior in this range of values and due to reduce the risk of injuries during the test avoiding exposure to high temperatures. In this sense, the maximum absolute temperature should be 70 °C. This maximum temperature avoids personal manipulation at higher temperature and damages due to HP contact with the aHB walls constructed with expanded polystyrene with polyurethane coating.

Defining a temperature gradient of 30 °C, different materials were tested. Defining the same temperature gradient for all tests maintains stable the lateral losses and the efficiency of the electronic system for all materials.

The results obtained using several materials are summarized in Table 3 and compared when are available with the standard values collected in the official catalog of constructive materials [14].

The materials analyzed with standard values maintain the error lower than 4.3% of the standard value. This error is enough for building applications.

Some additional materials without standard equivalent values were analyzed. The results have been used in different works to explore the use of new natural mixed materials with different concentrations to improve thermal resistivity. An example of this aHB application is shown in Figure 9 where different concentrations of iron (III) oxide mixed with lime mortar modify the U-value improving the thermal transmittance for building applications [45]. The lineal impact of iron oxide on the U-value was estimated using the aHB instrument saving cost and time with minimal accuracy loss.

The aHB instrument collects continuous sensor data which can be used to explore thermal features of sample under test at different test stages. Figure 10 shows the U-value evolutions for samples with different iron oxide concentrations. When the concentration increases, the U-value decreases. However, the ramp-up slope of transient stage is different for each concentration as well. In addition, the aHB instrument may be used to analyze thermal inertia behavior of materials related to the slope during the transient period.

The curves shown in Figure 10 highlight differences in the relation between iron oxide concentrations and the final U-value. The increment of concentration between 10% and 15% produces a wide gap between curves, which is not observed in the other concentration changes.

All this information can be estimated, analyzed, and considered to better understand the relationship between materials and thermal behavior using the new aHB instrument.

## 4. Discussion and Conclusions

Nowadays there are numerous commercial thermal characterization methodologies able to obtain the thermal transmittance of different building elements in a highly reliable way, using standards. However, such characterization methodologies impose strong impediments and restrictions both in the type of sample, measurement process and mainly total analysis time of a sample until the transmittance value is obtained.

The system proposed in this paper presents a hot-box system based on the ISO 8990:1994 standard that, although it does not meet the requirements set by the standard, is able to obtain a thermal transmittance value cheaper, because the aHB can be implemented using low-cost electronics and sensors, simpler, because the aHB instrument can be adapted to different sample sizes just by redesigning the insulating box, and faster, because it does not require a steady state to take the measurements and can obtain the U-value at lower thermal gradients, as reported in Figure 8.

It can, therefore, meet the needs of developers, builders and owners who want to know in detail, but without laboratory precision, the thermal characteristics of different construction elements, which is especially useful when using materials not present in official catalogs or new nature- based materials.

The proposed aHB can obtain results of thermal transmittance of a material in less than 18 h with adequate reliability, and without the need to keep the sample in quarantine to fix initial values, nor having to manipulate or substantially transform the geometry of the samples.

Although the thermal gradient is defined and digitally controlled, the heat-flux losses through the box as well as the flanking losses increase substantially when there are large external thermal variations. For this reason, it is recommended to use the aHB in indoor environments to maximize the linearity of the measurement.

## Figures and Tables

**Figure 1 sensors-23-01576-f001:**
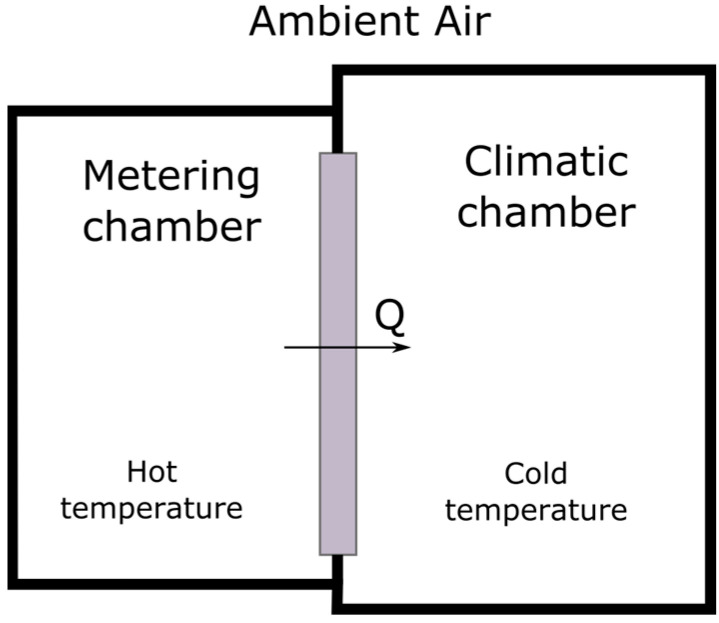
A simplified hot-box main parts schema.

**Figure 2 sensors-23-01576-f002:**
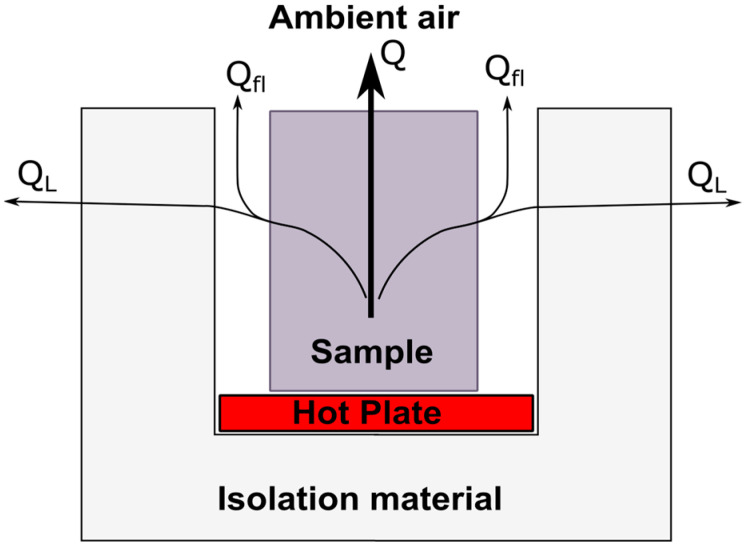
Ambient hot box schema with main thermal paths.

**Figure 3 sensors-23-01576-f003:**
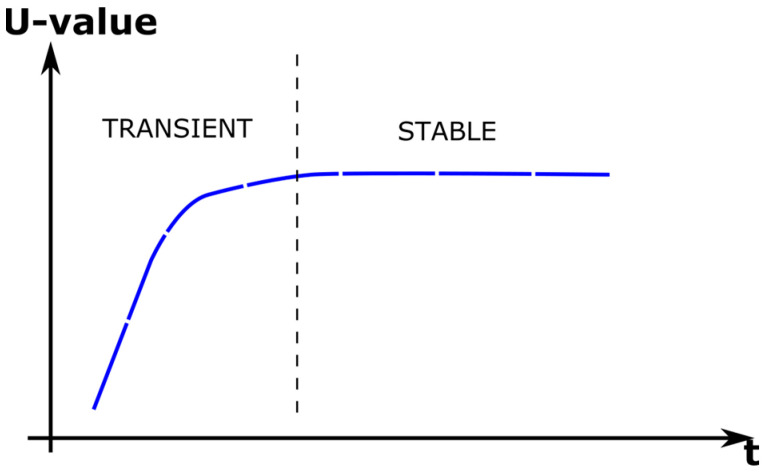
Schematic evolution of U-value with continuous thermal measurement.

**Figure 4 sensors-23-01576-f004:**
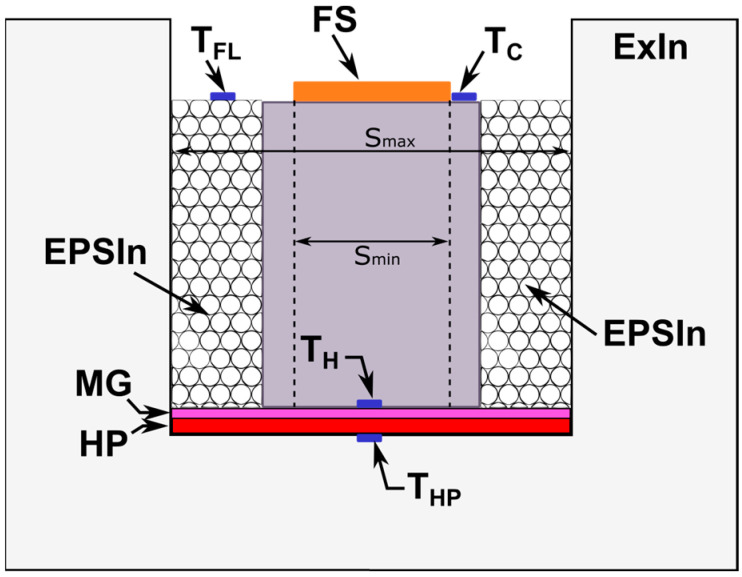
Ambient hot-box schema including the thermal insulation elements and sensors.

**Figure 5 sensors-23-01576-f005:**
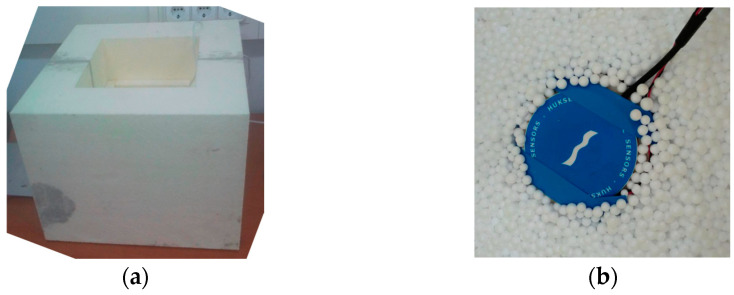
(**a**) External photography of the ambient hot-box and (**b**) detail of thermal flux sensor with EPSIn balls.

**Figure 6 sensors-23-01576-f006:**
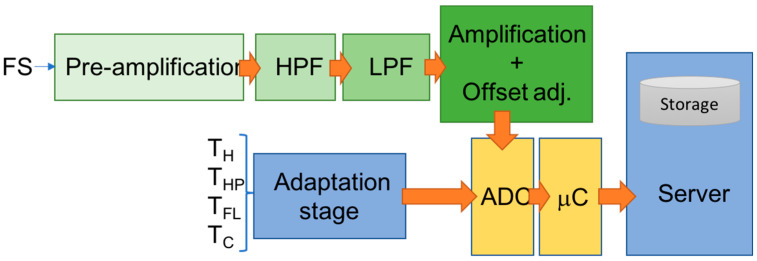
Outline of the electronic system responsible for aHB sensor adaptation and data signal acquisition.

**Figure 7 sensors-23-01576-f007:**
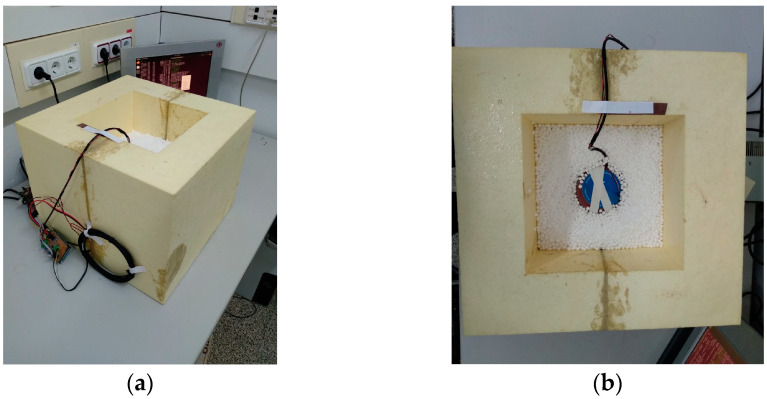
Pictures of aHB final implementation with electronic system board and data storage server: (**a**) lateral view of aHB with electronic boards; (**b**) top view of aHB with thermal flux on top of a material sample.

**Figure 8 sensors-23-01576-f008:**
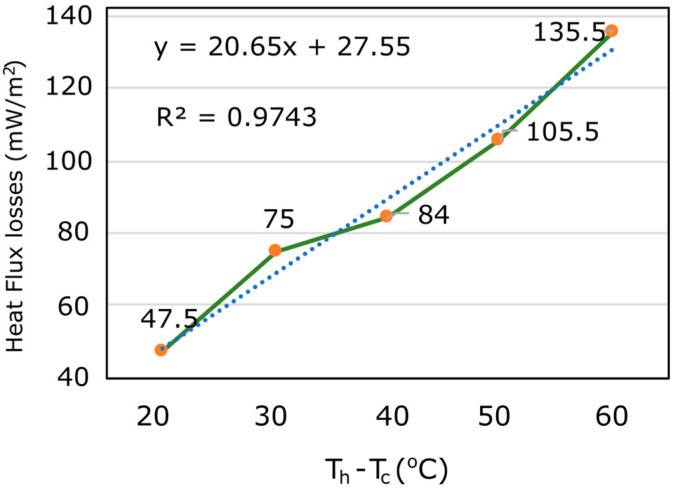
Lateral heat-flux losses as function of thermal gradient generated on a calibrated sample.

**Figure 9 sensors-23-01576-f009:**
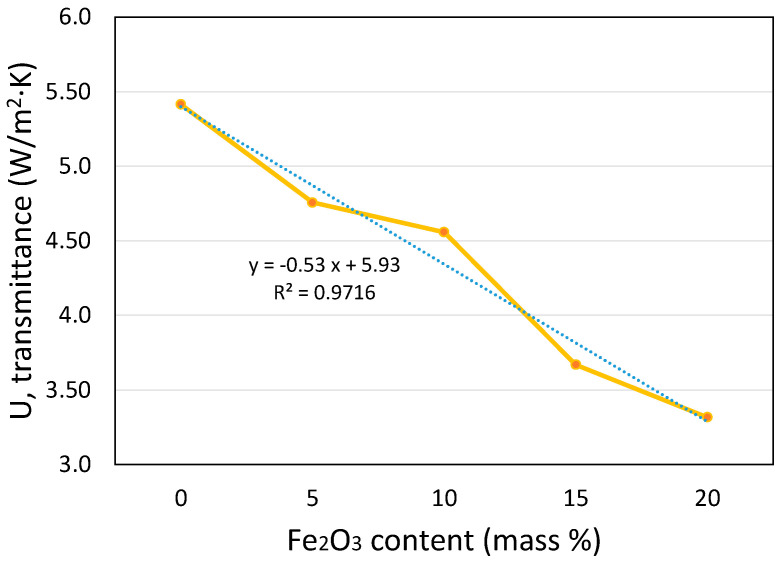
Transmittance of the samples of lime mortar with different iron (III) oxide content.

**Figure 10 sensors-23-01576-f010:**
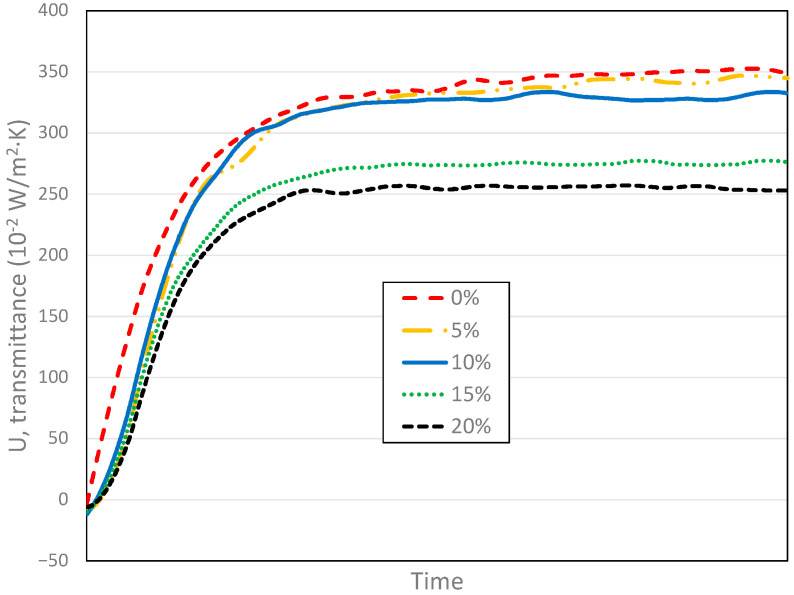
Continuous U-value evolution measured with aHB instrument using samples with different iron oxide concentrations.

**Table 1 sensors-23-01576-t001:** List of standards and their significance for the study.

Standard	Significance
EN ISO 8990 [31]	Thermal gradient determination, and thermal control of heat ambient
ASTM C1363-19 [32]	
GOST 26602.1-99 [41]	Heat flux measurement procedure
ISO 9869 [36]	Test duration, thermal resistance calculation and precision of calculation
ASTM C 1155 [37]	

**Table 2 sensors-23-01576-t002:** Heat-flux losses.

Location	% Loss
Bottom (*Qchw*/Qin)	3.62
Lower part of a lateral wall (*QL*/Qin)	2.83
Middle part of a lateral wall (*QL*/Qin)	2.14
Upper part of the lateral wall (*QL*/Qin)	1.75

**Table 3 sensors-23-01576-t003:** Some examples of measured materials.

Material	U (Standard) [14]	U Measured
Soft wood	0.15	0.17
Hard wood	0.23	0.202
Straw Bale	0.06	0.056
Limestone	1.7	1.773
Sandstone	1.5	1.515
Posidonia [44]	NA	0.045
Iron oxide plus lime [45]	NA	0.67
lime and cuttlefish bone mortar	NA	0.735

Note: NA—not available.

## Data Availability

Not applicable.

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
