# Peer review of "Ambient Hot Box: An Instrument for Thermal Characterization of Building Elements and Constructive Materials"

_sensors, 2023, doi:10.3390/s23031576_

Round 1

Reviewer 1 Report

The following comments should be incorporated into the manuscript to improve its quality.

1.In the abstract, paper presents, and paper offers sentences, could be combined as they are small sentences. 

2.Line 68 - … can be replaced by etc.

3.Lines 99,167,199,269,300 sentence vocabulary should be corrected.

4.Ambient heat box, Hot plate acronym defined in two places.

5.Many standards are used. All of them with their significance can be presented in a table form with just two columns, standard and significance.

6.In section 2.1.2, Smin,max used without their introduction/explanation directly.

7.Fig 8 – degree symbol is missing in the x-axis title

8.In the entire manuscript, symbol chosen for degree symbol is wrong. Choose the proper one without an underline.

9.In all graphs, outer border is no longer needed, and font should be black in colour instead of grey for normal representation.

10.In figure 10, same line style is used (for 10 and 20%) one could be changed. Because if printed black and white, no difference is observed

11.Units are missing in many graphs. Recheck.

Author Response

1.In the abstract, paper presents, and paper offers sentences, could be combined as they are small sentences.

Thank you for the comment. The abstract and introduction section have been reviewed considering this suggestion.

2.Line 68 - … can be replaced by etc.

Thank you for the comment. The document has been modified.

3.Lines 99,167,199,269,300 sentence vocabulary should be corrected.

Thank you very much for this comment. The sentences have been corrected improving the readability of the paper.

4.Ambient heat box, Hot plate acronym defined in two places.

The “Ambient heat box” is not used in the document, so the authors believe that the reviewer is referring to “Ambient Hot Box”. In that case, the document has been corrected.

“Hot Plate” acronym definition have been revised and the corrected.

5.Many standards are used. All of them with their significance can be presented in a table form with just two columns, standard and significance.

Thank you for the comment. The proposed table is included in the reviewed manuscript and the table captions have been updated.

6.In section 2.1.2, Smin,max used without their introduction/explanation directly.

The section has been modified to include a better explanation of the parameters.

7.Fig 8 – degree symbol is missing in the x-axis title

The degree symbol has been added to the x-axis title.

8.In the entire manuscript, symbol chosen for degree symbol is wrong. Choose the proper one without an underline.

Thank you for the comment. The complete manuscript is corrected with the correct degree symbol.

9.In all graphs, outer border is no longer needed, and font should be black in color instead of grey for normal representation.

All graphs have been changed according to reviewer’s comments

10.In figure 10, same line style is used (for 10 and 20%) one could be changed. Because if printed black and white, no difference is observed

Thank you for the comment. Figure 10 have been changed including a different style for each curve.

11.Units are missing in many graphs. Recheck.

Thank you for the comment. The graphs have been rechecked and Figures 9 and 10 are updated with the units.

Reviewer 2 Report

The article presents a new device and methodology for determining the thermal characteristics of a given material. The article has the correct form, it is legible, but the information contained in it raises some doubts: 1. The measurement is carried out in transient conditions; are there any temperature limits at which the test is conducted?; the question is dictated by the fact that depending on the difference in temperature, e.g. of heating plates (as in a plate apparatus) or two chambers (as in a heating box), the value of thermal conductivity or thermal resistance of the material may change. 2. How long does it take to stabilize the heat flux for different materials. (depending on their thermal parameters, e.g. heat capacity)? And therefore the measurement? 3. The heat flux recorder used for the measurement is very small, how does it affect the accuracy of the measurement? Moreover, can it be used to measure components or inhomogeneous materials? 4. The presented method resembles typical research methods in non-stationary conditions of the heat transfer coefficient U of the entire wall system, 5. How are heat transfer resistances assumed?

Author Response

  1. The measurement is carried out in transient conditions; are there any temperature limits at which the test is conducted?; the question is dictated by the fact that depending on the difference in temperature, e.g. of heating plates (as in a plate apparatus) or two chambers (as in a heating box), the value of thermal conductivity or thermal resistance of the material may change.

Thanks to the reviewer for the question. Indeed, the thermal resistance value may depend on the temperature difference applied to the sample under study. In this case and because it is a building oriented instrument, we have established that the highest gradient will be 45ºC. This would be the worst foreseeable case between the inside and outside of a house in Mediterranean latitudes. However, the instrument is capable of applying higher temperature gradients since the hot plate is capable of reaching up to 90ºC without affecting the integrity of the instrument.

A new paragraph has been included in the manuscript in section 3 regarding this question.

  1. How long does it take to stabilize the heat flux for different materials. (depending on their thermal parameters, e.g. heat capacity)? And therefore, the measurement?

Again, the reviewer has indicated a key detail in the use of the proposed instrument. Given that the scope of application to which the instrument is focused is construction materials, the authors have taken this into account to establish that the construction component with the highest thermal capacity and, therefore, that will require longer stabilization times of the temperature flow, will be the aggregates. In this sense, although the time for stabilization will depend on the materials of the sample, a minimum experiment time of 48h has been established. This maximizes the applicability of the instrument for the construction sector.

A new paragraph has been included in the manuscript regarding this question.

  1. The heat flux recorder used for the measurement is very small, how does it affect the accuracy of the measurement?

The construction of the box and the use of the EPSIn insulating filling material minimize the thermal dispersion of the sample by concentrating the transverse thermal flux under the area of the sensor used. In fact the minimum diameter of the sample under study (Smin) is defined from the diameter of the heat flux sensor. The effect on measurement accuracy is shown in the results with a maximum error of less than 4%, as stated in the article.

Moreover, can it be used to measure components or inhomogeneous materials?

The instrument can be used for the measurement of non-homogeneous materials, since it is not intended to analyze the interaction between each of the materials that form the sample under study, but the joint response produced, and above all the impact produced by the combination of materials in the stimation of the thermal resistance between the faces under measurement.

  1. The presented method resembles typical research methods in non-stationary conditions of the heat transfer coefficient U of the entire wall system,

The reviewer is correct in that the process is similar in both methodologies, given that a sample is exposed, either a portion of the building element in the case of the use of the exposed aHB, or the analysis of a complete wall in the U-value measurement systems. The main difference in the proposed instrument is that the second environment to which the sample is exposed is an uncontrolled environment. This simplification reduces stabilization times, but has the disadvantages described in the article that are overcome by a simple design and the measurement of reference thermal values.

  1. How are heat transfer resistances assumed?

The reviewer is asking about the heat transfer resistance due to roughness or impurities in the sample/sensor contact. In this study the heat transfer resistance are minimized because both, the sensor and the sample material are in direct contact,and the samples are treated to have both exposed faces are as flat as possible and provide as best contact as possible.